# HIERARCHICAL CORPUS ENCODER:
# FUSING GENERATIVE RETRIEVAL AND DENSE INDICES

## ABSTRACT

Generative retrieval employs sequence models for conditional generation of document IDs based on a query (DSI (Tay et al., 2022); NCI (Wang et al., 2022); *inter alia*). While this has led to improved performance in zero-shot retrieval, it is a challenge to support documents not seen during training. We identify the performance of generative retrieval lies in contrastive training between sibling nodes in a document hierarchy. This motivates our proposal, the *hierarchical corpus encoder* (HCE), which can be supported by traditional dense encoders. Our experiments show that HCE achieves superior results than generative retrieval models under both unsupervised zero-shot and supervised settings, while also allowing the easy addition and removal of documents to the index.

## 1 INTRODUCTION

Recent work on neural information retrieval (IR) has broadly fallen into two categories: *dense encoders*, and *generative retrieval*. For dense retrieval (Lee et al., 2019; Karpukhin et al., 2020; Izacard et al., 2022, *i.a.*), a fixed-dimension vector representation is created for each document by an encoder, and searching is performed with techniques such as approximate nearest neighbor (ANN) search or maximum inner product search (MIPS; Shrivastava & Li (2014)) with an external index (e.g., FAISS (Douze et al., 2024)). Such encoders are generally trained in a supervised fashion by learning to distinguish relevant documents (positive examples) against irrelevant documents (negative examples) given a query. This is known as contrastive training.

Generative retrieval (Tay et al., 2022; Wang et al., 2022, *i.a.*) takes a different approach: directly outputting the identifier of the document with an encoder-decoder sequence model without an explicit vector representation for each document. Purported advantages include (a) there is no need to deploy an explicit MIPS index; and (b) better performance in unsupervised settings when adapted to a new corpus without any labeled training data. However, they also suffer from known problems such as the addition of new documents requires continued training, which is both computationally expensive and is prone to catastrophic forgetting (Kishore et al., 2023).

Here we examine the innovations of generative retrieval and identify the key important distinction with dense retrieval approaches to date: *tiered hierarchical negative samples*. This motivates our proposal, the *hierarchical corpus encoder* (HCE). At training time, positive samples are contrasted against siblings on a document hierarchy as negative samples, mimicking the loss employed in generative retrieval (§3). At test time, retrieval falls back to MIPS with an external index. HCE provides the best of both worlds: (1) zero-shot adaptation to new domains; and (2) efficient addition and removal to the index without fine-tuning.

Our experimental results demonstrate that HCE achieves superior performance over a variety of popular dense and generative retrieval methods under both supervised and unsupervised scenarios, illustrating the effectiveness of HCE's modeling of the document set as a hierarchy.

## 2 BACKGROUND

**Dense Retrieval**  In the dense retrieval paradigm of information retrieval, one seeks to learn an *encoder* $\mathbf{F} : V^* \rightarrow \mathbb{R}^n$ that maps a string of tokens (from vocabulary set $V$) to a point in a $n$-dimensional vector space. Retrieval can then be performed in by some nearest neighbor search (NNS) or maximum inner product search (MIPS) in this space $\mathbb{R}^n$. A common instantiation of this encoder

is a Transformer with a pooling operation on top (Izacard et al., 2022), followed by an optional normalization step (that makes the norm of the vector 1, as is done in Ni et al. (2022)).

Under the condition where query-document relevance judgments are present, it is common practice to train such an encoder with a *constrastive loss* (Sohn, 2016), where the model is trained to discriminate positive candidates $d^+$ that are relevant to the query $q$ from irrelevant negative candidates $d^- \in D^-$. We denote the vector representation of a query $q$ as $\mathbf{q} = \mathbf{F}(q)$, and similarly $\mathbf{d} = \mathbf{F}(d)$. For a set of documents $D$, we denote $\mathbf{D} = \{\mathbf{F}(d)\}_{d \in D}$, which is a set of vectors. A common form of the loss is

$$\mathcal{L}_C(\mathbf{q}, \mathbf{d}^+, \mathbf{D}^-) = -\log \frac{\exp S(\mathbf{q}, \mathbf{d}^+)}{\exp S(\mathbf{q}, \mathbf{d}^+) + \sum_{\mathbf{d}^- \in \mathbf{D}^-} \exp S(\mathbf{q}, \mathbf{d}^-)} \ , \tag{1}$$

where $S(\mathbf{q}, \mathbf{d})$ is the scoring function between vectors. This scoring function is usually just an inner product (optionally scaled by a temperature $\tau$) between vector embeddings $S(\mathbf{q}, \mathbf{d}) = \mathbf{q} \cdot \mathbf{d}/\tau$, or a normalized version $S(q, d) = \frac{\mathbf{q} \cdot \mathbf{d}}{\tau \cdot \|\mathbf{q}\| \cdot \|\mathbf{d}\|}$ where cosine similarity is computed.

This general framework is first pioneered by Lee et al. (2019) and Karpukhin et al. (2020) for dense retrieval in NLP, where it is successfully applied to open-domain question answering.

**Generative Retrieval** Generative retrieval is a new paradigm that directly generates document identifiers by the model's *parametric memory*, without using an external index. Specifically, one first assigns a unique string-valued identifier to each document, and when decoding, constrain the decoding process so that its output falls within the set of unique identifiers. Top-$k$ retrieval is done by using beam search in the decoding process. Such method removes the need for any external index, making the whole memory of the corpus *parametrized by* a neural network.

The earliest work in this thread is GENRE (De Cao et al., 2021), where the correct entity name (e.g. Wikipedia article title) is generated through a sequence-to-sequence model for entity linking. DSI (Tay et al., 2022) and NCI (Wang et al., 2022) applied the generative retrieval approach to *ad hoc* document retrieval. GENRET (Sun et al., 2023) learns the document IDs without the initial preprocessing step. Such methods suffer from two drawbacks: (a) *scalability*: challenging to scale to massive scale of documents since all memory is parametrized in the model, absent of external storage; (b) *extensibility*: hard to expand the document set if new documents are going to be indexed, since it is usually pre-trained on a static collection of documents.

Adding new documents in generative retrieval usually requires continued training. There are methods proposed to alleviate the problem: DSI++ (Mehta et al., 2023) sought to mitigate catastrophic forgetting for continued training; IncDSI (Kishore et al., 2023) proposed a constrained optimization method to find optimal vectors for new documents, but it only applies to the *atomic* version of DSI.

## 3 A CLOSER LOOK AT GENERATIVE RETRIEVAL

Generative retrieval seems to be a departure from traditional contrastive learning, but upon scrutiny, we found that the underlying loss is similar in many ways. Here we take a closer look at arguably the seminal work that started the research thread on generative retrieval for documents: the *differentiable search index* (DSI; Tay et al., 2022). There were two proposed versions of DSI, namely the *atomic*-ID version and the *hierarchical* version. Later work that followed (e.g. NCI; Wang et al. (2022)) prioritized the hierarchical version.

**Atomic Version** In the *atomic* version of DSI, each document's ID is added to the vocabulary of the Seq2Seq decoder, and the decoder is run exactly 1 step to generate the correct document ID. The Seq2Seq negative log-likelihood loss would be[1]

$$\mathcal{L}_{\text{GR-atom}}(q, d^+) = -\log\left(\text{softmax}(\mathbf{W}_{\text{LMHead}} \cdot \mathbf{s})[d^+]\right). \tag{2}$$

Note here $\mathbf{s} \in \mathbb{R}^{n_{\text{dec}}}$ is the decoder output state (Figure 1 Left), and $\mathbf{W}_{\text{LMHead}}$ is the weight matrix of the *language model head* of the decoder, and $\mathbf{x}[d^+]$ selects the probability term from vector $\mathbf{x}$ that corresponds to the document $d^+$. Since the token set of this decoder is the *document ID set*,

---

[1] Bias term omitted for more succinct exposition, as one can always rewrite $(\mathbf{w}^T\mathbf{x} + b)$ as $(\mathbf{w}, b)^T(\mathbf{x}, 1)$.

$$\mathbf{W}_{\text{LMHead}} = \left[ \mathbf{v}_d^{\text{T}} \right]_{d \in \mathcal{D}} \in \mathbb{R}^{|\mathcal{D}| \times n_{\text{dec}}}, \text{ and the logits } (\mathbf{W}_{\text{LMHead}} \cdot \mathbf{s}) \in \mathbb{R}^{|\mathcal{D}|}. \text{ Expanding Equation 2:}$$

$$\mathcal{L}_{\text{GR-atom}}(q, d^+) = -\log \frac{\exp \mathbf{s} \cdot \mathbf{v}_{d^+}}{\sum_{d \in \mathcal{D}} \exp \mathbf{s} \cdot \mathbf{v}_{d^+}} = \mathcal{L}_C \left( \mathbf{s}, \mathbf{v}_{d^+}, \{\mathbf{v}_{d^-}\}_{d^- \in \mathcal{D} \setminus \{d^+\}} \right). \tag{3}$$

Thus we see that in the *atomic* version of generative retrieval,

- a fixed-length vector $\mathbf{v}_d$ is actually trained for each document $d$, in the form of rows in the weight matrix of the language model head;
- a fixed-length vector is actually created for the query, in the form of the decoder state vector $\mathbf{s}$;
- and the decoding process of $\arg\max_{d \in \mathcal{D}} \exp \mathbf{s} \cdot \mathbf{v}_d$ is in fact maximum inner product search (MIPS).

Hence the atomic version of generative retrieval can be considered as a form of contrastive learning, where the positive document is contrasted with *all* other documents in the corpus, with the embedding for all documents saved as *parametric memory* in $\mathbf{W}_{\text{LMHead}}$ (Figure 1 Left). These embeddings are updated under gradient descent for every training iteration. This is different from contrastive learning in dense retrievers, where usually a small set of negative samples are *sampled* from the corpus.

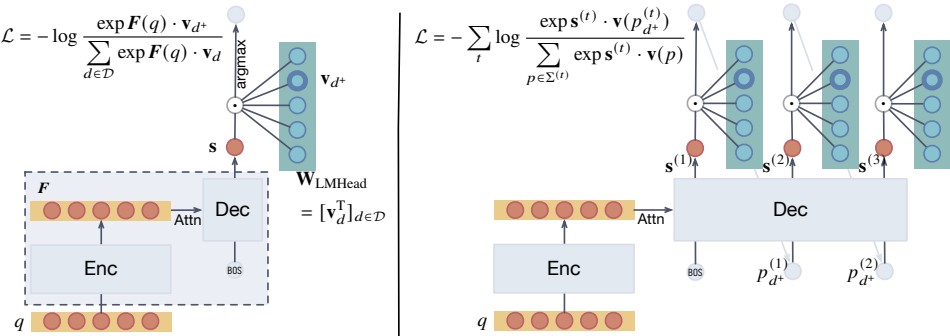

Figure 1: Left: DSI with atomic IDs. Right: DSI with a document hierarchy.

**Hierarchical Version**   The atomic version above obviously does not scale efficiently beyond hundreds of thousands of candidate documents, as the size of its parametric memory would scale linearly with respect to the number of documents. Therefore the authors of DSI proposed a hierarchical method that limits the number of tokens that the decoder can generate. In the hierarchical version of DSI, a hierarchy of documents is computed before training via divisive $k$-means clustering. The set of documents are arranged as leaves in a tree, where intermediate nodes are clustering centroids. In this tree $\mathcal{T}$, each document $d$ is assigned a path from root $\mathbf{p}_d = (p_d^{(1)}, \cdots, p_d^{|\mathbf{p}_d|})$ (Figure 1 Right).

The DSI decoder is expected to output this path $\mathbf{p}_d$ as the sequence output. Since DSI is trained with a sequence objective with a softmax head, this sequence loss can therefore be expressed as

$$\mathcal{L}_{\text{GR-hier}}(q, d^+) = -\sum_{t=1}^{|\mathbf{p}_{d^+}|} \log \frac{\exp \mathbf{s}^{(t)} \cdot \mathbf{v}(p_{d^+}^{(t)})}{\sum_{p \in \Sigma^{(t)}} \exp \mathbf{s}^{(t)} \cdot \mathbf{v}(p)} = \sum_{t=1}^{|\mathbf{p}_{d^+}|} \mathcal{L}_C \left( \mathbf{s}^{(t)}, \mathbf{v}(p_{d^+}^{(t)}), \{\mathbf{v}_p\}_{p \in \Sigma^{(t)}} \right), \tag{4}$$

where $t$ is the decoder step, $\mathbf{s}^{(t)}$ is the decoder state at step $t$, and $\Sigma^{(t)}$ is the set of symbols allowed on depth $t$ of the hierarchy. Here we see that the hierarchical version of generative retrieval performs contrastive learning at each step in the decoding process: at each step $t$, the decoder state $\mathbf{s}^{(t)}$, acting as a *query*, is matched with all possible tokens $\Sigma^{(t)}$ at this step $t$: the correct action at this step $p_{d^+}^{(t)}$ is contrasted against all other steps. In essence, DSI is taking *tiered hierarchical negative samples*.

It can be seen that the loss function of generative retrieval essentially consists of one (under the atomic version) or multiple (under the hierarchical version) steps of contrastive loss, with the decoder state at each step functioning as the query vector. We make the observation that *contrastive loss* is a sampled estimation of the full log-likelihood loss; and the hierarchical version is a computationally efficient way of working with a very large candidate set. Indeed, the same idea has been explored in

**Algorithm 1** HIERAGGCLUSTER

**Require:** vectors $\mathbf{v}_i \in \mathbb{R}^n$, $1 \le i \le |\mathcal{D}|$
**Require:** branching factor $b$
  $t \leftarrow \lceil \log_b |\mathcal{D}| \rceil$
  $T^{(t)} \leftarrow \{\text{TREE}(\mathbf{v}_i, \varnothing)\}$   ▷ *A forest of leaf nodes*
  $K \leftarrow \lceil |\mathcal{D}|/b \rceil$
  **while** $t > 0$ **do**
    $T^{(t-1)} \leftarrow \text{SPHKMEANS}(T^{(t)}, K)$
             ▷ *Clusters into a forest of subtrees*
    $t \leftarrow t - 1$
    $K \leftarrow \lceil K/b \rceil$
  **end while**
  **return** $T^{(0)}$   ▷ *Returns a tree with a single root*

**Algorithm 2** SPHKMEANS

**Require:** vectors $\mathbf{v}_i \in \mathbb{R}^n$, $1 \le i \le |\mathcal{D}|$
**Require:** number of clusters $K$
  $\forall i,\ a_i \sim \text{Unif}\{1, \cdots, K\}$   ▷ *Random init*
  **while** $a_i$ not converged **do**
$$\forall k,\ \mathbf{c}_k \leftarrow \frac{\sum_{i:a_i=k} \mathbf{v}_i}{\left\| \sum_{i:a_i=k} \mathbf{v}_i \right\|} \qquad \triangleright \textit{E step}$$
    $\forall i,\ a_i \leftarrow \arg\max_k \mathbf{v}_i \cdot \mathbf{c}_k$   ▷ *M step*
  **end while**
  **return** $\left\{ \text{TREE}(\mathbf{c}_k, \{i : \mathbf{v}_i\}_{a_i=k}) \right\}_{1 \le k \le K}$
             ▷ *Returns a forest of clusters*
  ▷ *TREE(r, C) is a tree with root r and children C*

many applications of NLP. For example, in the early days of neural language modeling, Morin & Bengio (2005) proposed a *hierarchical softmax* function to speed up the softmax computation over a large vocabulary set. This was then adopted in Word2Vec (Mikolov et al., 2013).

In generative retrieval, decoding is done in a coarse-to-fine manner with constrained beam search. This is akin to the approach of Chen et al. (2020b), who classified textual entity mentions into a constraining type hierarchy. Hierarchical decoding is also reminiscent of indexing algorithms such as hierarchical $k$-means trees (Nistér & Stewénius, 2006; Muja & Lowe, 2009): first build a tree of samples based on clustering as index, and when decoding, apply beam search over this tree.

With this understanding of generative retrieval, we translate the innovations to a dense retriever.

# 4 MODEL

We propose HCE, which jointly learns an encoder $\mathbf{F} : V^* \rightarrow \mathbb{R}^n$ (the $n$-dimensional hypersphere $\mathcal{S}^{n-1} \subset \mathbb{R}^n$ if normalized) with a hierarchical tree $\mathcal{T}$ of the document set, with a novel loss that takes the hierarchy of documents into account.

## 4.1 LEARNING WITH DOCUMENT HIERARCHY

**Clustering**   Given an initial encoder $\mathbf{F}_0$, we can compute all embeddings in the document set $\mathcal{D}$: $\{\mathbf{F}_0(d)\}_{d \in \mathcal{D}}$. We perform an agglomerative version of hierarchical clustering (Algorithm 1) on these vectors to form a tree. This differs from DSI, where divisive clustering is performed. The reason is that we keep the path from the root the same length for all documents (see Figure 2), and easier parallelization on GPUs.

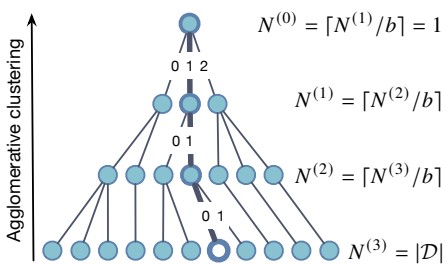

Figure 2: A document hierarchy with depth 3.

Starting with $|\mathcal{D}|$ vectors for the entire corpus, we perform spherical $K$-means clustering,[2] where $K = \lceil |\mathcal{D}|/b \rceil$. Here $b$ is a *branching factor*. We recurse until $K = 1$, when all clusters are collected into a single root node. Note that for each tree node there is no guarantee that it has exactly $b$ children, and $b$ can be understood as the expected number of vectors in each cluster (see Figure 2). For each clustering step, the spherical $K$-means clustering (Dhillon & Modha, 2001) is used (Algorithm 2).

The resulting hierarchical tree $\mathcal{T} = T^{(0)}$ has depth $L = \lceil \log_b |\mathcal{D}| \rceil$, so that each document $d$ can be encoded as a fixed-length path $\mathbf{p}_d = (p_d^{(1)}, \cdots, p_d^{(L)})$ from root (see Figure 2, where the highlighted leaf node has its path to root $\mathbf{p} = (1, 1, 0)$ bolded). Each prefix $(p^{(1)}, \cdots, p^{(l)})$ $(l < L)$ of this path points to a *nonterminal* node $c$ of this tree, and corresponds to a centroid from the hierarchical clustering process. We denote the vector of the centroid as $\mathbf{c}(p^{(1)}, \cdots, p^{(l)}) \in \mathbb{R}^n$.

**Hierarchy-aware Loss**   Provided that the hierarchy is constructed, we adapt the sequence decoding loss ($\mathcal{L}_{\text{GR-hier}}$, Equation 4) in generative retrieval to our encoder-based method. Recall that $\mathcal{L}_{\text{GR-hier}}$

---

[2] Spherical $K$-means is used instead of normal $K$-means since most pretrained retriever maximize inner product or cosine similarity between relevant query-document pairs, instead of minimizing $L_2$ distance.

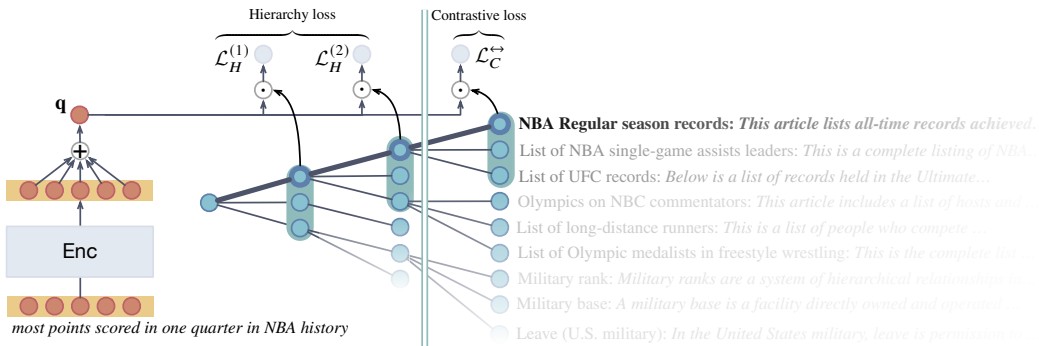

Figure 3: Illustration of HCE training, where the query is contrasted with tiered negative samples. Query and documents here are taken from the NQ320k dataset (Kwiatkowski et al., 2019).

contrasts the vector of the positive path against the vector of the negative paths on each layer of the hierarchy, given the decoder state at each step $\mathbf{s}^{(t)}$ as the query vector. We make two modifications:

- Since there is no decoder in our case, the query vector *stays the same across steps*: it will always be the vector embedding $\mathbf{q} = \mathbf{F}(q)$ of the query $q$ across levels.

- Vectors for intermediate nodes are the *centroid vectors of the prefixes* from $K$-means clustering.

As such, given query $q$ and its relevant document $d^+$, at step $t$ on hierarchy $\mathcal{T}$, we contrast the positive prefix $(p_{d^+}^{(0)}, \cdots, p_{d^+}^{(t)})$ against *all its siblings* $\left\{ (p_{d^+}^{(0)}, \cdots, p_{d^+}^{(t-1)}, p^-) \,\middle|\, p^- \neq p_{d^+}^{(t)} \right\}$ (Figure 3):

$$\mathcal{L}_H^{(t)}(q, d^+) = \mathcal{L}_C \left( \mathbf{q}, \ \mathbf{c}(p_{d^+}^{(0)}, \cdots, p_{d^+}^{(t)}), \ \left\{ \mathbf{c}(p_{d^+}^{(0)}, \cdots, p_{d^+}^{(t-1)}, p^-) \,\middle|\, p^- \neq p_{d^+}^{(t)} \right\} \right) \tag{5}$$

Naturally we could do this for each layer of the hierarchy: $\mathcal{L}_H(q, d^+) = \sum_{t=1}^{L} \mathcal{L}_H^{(t)}(q, d^+)$, taking *tiered hierarchical negative samples*. However, doing this for the full hierarchy is not memory-efficient: it requires storing the embedding of all documents (leaf nodes of the hierarchy) as parameters! This gets us back to the situation of generative retrieval with atomic IDs (§3). We adopt a simple solution: retain the vectors of the first $M (M < L)$ layers of the $L$ layers in memory. For the first $M$ layers, we apply the hierarchy loss in Equation 5; for the last $(L - M)$ layers, we fall back to constrastive loss, where negative samples are sampled within the children of the prefix (Figure 3):

$$\mathcal{L}_{\text{HCE}}(q, d^+) = \underbrace{\sum_{t=1}^{M} \mathcal{L}_H^{(t)}(q, d^+)}_{\text{Hierarchy loss}} + \underbrace{\sum_{t=M+1}^{L} \mathcal{L}_C^{\leftrightarrow} \left( q, d^+, \text{Sample}_{n_{\text{NS}}} \left( p_{d^+}^{(1)}, \cdots, p_{d^+}^{(t)} \right) \setminus \{d^+\} \right)}_{\text{Contrastive loss with negative samples}}. \tag{6}$$

Here $\mathcal{L}_C^{\leftrightarrow}$ is a bi-directional contrastive loss elaborated below, and $\text{Sample}_{n_{\text{NS}}}(p)$ samples $n_{\text{NS}}$ documents that are the children of the prefix $p$. In our experiments we take $n_{\text{NS}} = 4$.

**Contrastive Loss with Tricks** It has been shown that the *in-batch negative trick* (Henderson et al., 2017; Chen et al., 2020a) improves the performance by including positive documents for other queries $j \neq i$ in the same *mini-batch*[3] $B$ as negative documents for query $i$. Additionally, Ni et al. (2022) shows that one can do a bi-directional loss: viewing the candidate as the query and the queries in the batch as candidates. We combine these tricks. For details, see Appendix A.

### 4.2 TRAINING UNDER DIFFERENT SCENARIOS

Our proposed method HCE can be used in both supervised training where query-document relevance judgments are present; and in zero-shot cases where query dataset is absent.

**Supervised Training** Given a training set of $\mathcal{S}_{\text{train}} = \{(q, d^+)\}$ (there is no need for labeled irrelevant samples) and a corpus $\mathcal{D}$, we simply optimize the HCE loss (Equation 6) proposed above.

---

[3] Batch size can be increased by the number of GPUs by using the `AllGather` operation, e.g. in `NCCL`.

**Unsupervised Training** In this scenario only the corpus $\mathcal{D}$ is present, and our goal of this training step is to align the encoder's output distribution with the distribution of the corpus.

The goal is creating a pseudo-query for each document. We experimented with two approaches:

- **Inverse Cloze Task** (ICT; Lee et al. (2019)): A random segment $\tilde{q}$ of text $d^+$ is used as a pseudo-query: $\mathcal{L}_U(d^+) = \mathcal{L}_{\text{HCE}}(\tilde{\mathbf{q}}, \mathbf{d}^+)$. In NCI (Wang et al., 2022) it is referred to as *document-as-query*, and we follow their choice of taking a random span of 64 tokens;

- **SimCSE** (Gao et al., 2021): The text $d^+$ is encoded again with a different randomized dropout mask, yielding a different encoding $\tilde{\mathbf{d}}^+$ from the original $\mathbf{d}^+$. Then the variant $\tilde{\mathbf{d}}^+$ is used as the query vector: $\mathcal{L}_U(d^+) = \mathcal{L}_{\text{HCE}}(\tilde{\mathbf{d}}^+, \mathbf{d}^+)$.

We found that these two methods perform similarly. For all experiments described, ICT is used.

**Joint Training** Additionally, the supervised and the unsupervised objective can be *jointly* optimized:

$$\mathcal{L}_J(\mathcal{S}_{\text{train}}, \mathcal{D}) = \mathbb{E}_{(q,d^+)\sim\mathcal{S}_{\text{train}}} \left[ \mathcal{L}_S(q, d^+) \right] + \alpha \cdot \mathbb{E}_{d\sim\mathcal{D}} \left[ \mathcal{L}_U(d) \right]. \tag{7}$$

This achieves the two goals outlined in Wang & Isola (2020): (a) aligning the query with its relevant document; (b) ensures that the distribution of the corpus is nicely distributed.

**Co-Training of Encoder and Hierarchy** Finally, we introduce a co-training approach for jointly optimizing the encoder and the hierarchy. In existing generative retrievers (e.g. DSI, NCI), the construction of the hierarchy is a preprocessing step, and is usually computed from another encoder.[4] We seek to jointly optimize these in an EM-style (coordinate descent with alternating maximization) co-training setup (Alg. 3).

In essence, after each epoch, if the metric on the validation set increases (i.e., a better representation of the corpus is obtained), a re-clustering of the corpus will be triggered. We stress that after training completes, the hierarchy $\mathcal{T}$ can be *safely discarded* as only the encoder $\mathbf{F}$ is needed for downstream indexing and retrieval.

---

**Algorithm 3** EM-STYLE-TRAIN

---

**Require:** Training dataset $\mathcal{S}_{\text{train}}$
**Require:** Validation dataset $\mathcal{S}_{\text{dev}}$
**Require:** Document collection $\mathcal{D}$
**Require:** Initial model checkpoint $\mathbf{F}_0$
  $\mathbf{F} \leftarrow \mathbf{F}_0$
  $\mathcal{T} \leftarrow \text{HIERAGGCLUSTER}(\{\mathbf{F}_0(d)\}_{d\in\mathcal{D}})$
  $m \leftarrow \text{METRIC}(\mathbf{F}, \mathcal{S}_{\text{dev}}, \mathcal{D})$   ▷ *Some metric on* $\mathbf{F}_0$
  **while** early stopping criteria not met **do**
    $\mathbf{F}' \leftarrow \text{OPTIMIZE}(\mathbf{F}, \mathcal{T}, \mathcal{S}_{\text{train}}, \mathcal{D})$
    $m' \leftarrow \text{METRIC}(\mathbf{F}', \mathcal{S}_{\text{dev}}, \mathcal{D})$
    **if** $m' > m$ **then**   ▷ *A better representation found*
      $\mathcal{T} \leftarrow \text{HIERAGGCLUSTER}(\mathbf{F}'(d)_{d\in\mathcal{D}})$
      $m \leftarrow m'$
    **end if**
    $\mathbf{F} \leftarrow \mathbf{F}'$
  **end while**
  **return** $\mathbf{F}$

---

## 5 EXPERIMENTS & DISCUSSIONS

**Setup** We start from the pre-trained dense encoder GTR[5] (Ni et al., 2022) as both a baseline and the initial checkpoint where we continue training, as this is also used as a baseline in DSI (Tay et al., 2022). Additionally GTR, DSI, and NCI are all fine-tuned versions of T5, facilitating fair comparison.

We mostly follow the configurations as specified in GTR, where the temperature for contrastive loss is set as $\tau = 0.01$, with the AdaFactor optimizer (Shazeer & Stern, 2018) with linear decay in accordance with GTR's hyperparameters.[6] We use machines with 8× NVidia A100 GPUs with `bfloat16` precision, with total batch size across GPUs 512 (for the ArguAna dataset, 192). No experiment ran for more than 1 day.

**HCE-{U, S, J}** stands for unsupervised, supervised, and joint training respectively. For HCE-J, the loss weight for unsupervised learning in Equation 7 is set at $\alpha = 0.5$.

---

[4] In both DSI and NCI, document vectors used to generate the hierarchy is computed with BERT-`base` whereas their model is based on T5, causing a discrepancy. HCE uses the same underlying model.

[5] GTR normalizes the embeddings so that their $L_2$ norm is 1, thus mapping into the hypersphere $\mathcal{S}^{n-1}$.

[6] A minor difference is that we employ an initial learning rate $\eta = 10^{-4}$, since we found that the original $\eta = 10^{-3}$ causes unstable oscillation under unsupervised learning.

## 5.1 SUPERVISED SETTING: NQ320K & TRIVIAQA

**Datasets**  We employ two datasets commonly used in prior work for supervised retrieval training: NaturalQuestions (Kwiatkowski et al., 2019) and TriviaQA (Joshi et al., 2017). NaturalQuestions collects real-user queries from Google and each query is paired with an answer span from a Wikipedia article. A common version of the dataset is NQ320k, where there are 320k query-passage pairs. TriviaQA is a reading comprehension dataset whose candidate passages also come from Wikipedia. We follow the preprocessing scripts in NCI (Wang et al., 2022). For details see Appendix B.

**Baselines and Metrics**  We include the following methods as baselines, grouped by their categories:

- **Sparse retrieval:** BM25 (Robertson et al., 1994) and a version with augmented generated queries by DocT5Query (Nogueira & Lin, 2019).
- **Pre-trained dense retrievers:** These are dense retrievers pre-trained or fine-tuned with other datasets such as MS MARCO (Nguyen et al., 2016), but not fine-tuned on the datasets here. These include Contriever fine-tuned on MS MARCO (Izacard et al., 2022) (based on BERT) and GTR (based on T5).
- **Fine-tuned dense retrievers:** Dense retrievers fine-tuned on NQ320k or TriviaQA. These include DPR (Karpukhin et al., 2020) and ANCE (Xiong et al., 2021). We also fine-tune our initial checkpoint, GTR, with ANCE-style contrastive training only (without hierarchy, denoted with "+CT") to illustrate the gain of our novel hierarchy loss. Our full HCE also falls into this category.
- **Generative retrievers:** DSI (Tay et al., 2022); NCI (Wang et al., 2022); SEAL (Bevilacqua et al., 2022) where substrings are generated as document IDs; and GENRET (Sun et al., 2023), the state-of-the-art generative retriever that learns document IDs without clustering as preprocessing.

We use the NIST `trec_eval`[7] tool, reporting the union of the common metrics employed in prior work, including recall at various cutoff $k$ (R@$k$), and mean reciprocal rank (MRR).[8]

**Hyperparameters**  Learning rate, temperature, etc. are set in the prior section. Here the only parameter we tune is the branching factor $b$, which impacts the performance (see analysis below).

**Query Generation**  In some of the baseline methods, query generation (QG) is applied to the document set — these retrievers are marked with "+QG" in Table 1. In all of these retrievers, DocT5Query (Nogueira & Lin, 2019) fine-tuned by MS MARCO (Nguyen et al., 2016) is employed. For sparse retrievers, generated queries are appended to the document to be indexed. For dense & generative retrievers, generated queries and corresponding documents are added to the training set.

**Results**  Jointly trained HCE outperformed all dense and generative retrieval baselines across both datasets (Table 1). This demonstrates the effectiveness of HCE's learned vector representations.[9] HCE is especially good at recall@$k$ where $k > 5$, with a 6–7% boost against NCI and 2% gain against GENRET under NQ320k, and a 2% boost against NCI for TriviaQA.

HCE's performance gain is less in recall@$k$ when $k$ is small, as compared to a large $k$. Our hypothesis is that when discriminating between similar documents under the same prefix in the hierarchy, HCE uses a negative-sampling approach for computational efficiency, resulting in less negative samples than the full set under the prefix being trained. This in turn causes good performance on the first $M$ levels of the hierarchy but not as good at the last layer of the hierarchy.

**Discussion: Ablation Studies**  Here we remove certain components from our best setup to see the contribution of each component. From Table 1, we see that by removing query generation or hierarchy co-training (Figure 3), the performance of R@1 drops about 1%, while for recall at $k >= 10$ the results are largely the same. Comparing HCE with GTR+CT we found that our novel hierarchy-aware loss improves the performance of R@1 about 4% across datasets.

---

[7] https://github.com/usnistgov/trec_eval.

[8] In some prior work, hit@$k$ (a.k.a. success@$k$) is reported, which is defined to be the ratio of queries where at least 1 relevant document is found within the top-$k$ retrieved set. Note that for NQ320k, each query is paired with at most 1 relevant document, thus recall@$k$ = hit@$k$ for NQ320k. However, for TriviaQA, there may be more than 1 relevant document for a query, hence recall@$k$ ≠ hit@$k$.

[9] Note that HCE is an *encoder-only* model: it has only half of the parameters of DSI and NCI models.

| Model | NQ320k | | | | | TriviaQA | | |
|---|---|---|---|---|---|---|---|---|
| | R@1 | R@5 | R@10 | R@100 | MRR | R@5 | R@20 | R@100 |
| **Sparse retrievers** | | | | | | | | |
| BM25 | 0.297 | 0.508 | 0.603 | 0.821 | 0.402 | 0.569 | 0.695 | 0.802 |
| BM25 (+QG) | 0.380 | 0.591 | 0.693 | 0.861 | 0.489 | 0.597 | 0.721 | 0.827 |
| **Pre-trained dense retrievers** | | | | | | | | |
| Contriever | 0.436 | 0.717 | 0.796 | 0.944 | 0.561 | 0.565 | 0.706 | 0.832 |
| Contriever-MSMARCO | 0.597 | 0.840 | 0.902 | 0.973 | 0.705 | 0.645 | 0.762 | 0.869 |
| GTR (`base`) | 0.560 | 0.792 | 0.844 | 0.937 | 0.662 | 0.610 | 0.724 | 0.832 |
| GTR (`large`) | 0.565 | 0.801 | 0.854 | 0.944 | 0.677 | 0.625 | 0.742 | 0.852 |
| **Supervisedly trained dense retrievers** | | | | | | | | |
| DPR | *0.463* | *0.695* | *0.756* | *0.909* | *0.569* | *0.512* | *0.622* | *0.741* |
| ANCE | 0.526 | | 0.804 | 0.913 | 0.628 | | 0.803 | 0.853 |
| GTR (`large`), +CT | 0.622 | 0.869 | 0.926 | 0.968 | 0.735 | 0.747 | 0.876 | 0.948 |
| GTR (`large`), +CT, +QG | 0.672 | 0.896 | 0.927 | 0.972 | 0.775 | 0.763 | 0.887 | 0.953 |
| **Generative retrievers** | | | | | | | | |
| DSI (`base`) | 0.274 | | 0.566 | | | | | |
| DSI (`large`) | 0.356 | | 0.626 | | | | | |
| NCI (`base`) | 0.602 | *0.766* | 0.802 | 0.909 | 0.679 | | | |
| NCI (`base`), +QG | 0.659 | *0.821* | 0.852 | 0.924 | 0.731 | 0.782 | 0.873 | 0.928 |
| NCI (`large`), +QG | 0.662 | *0.817* | 0.853 | 0.925 | 0.734 | 0.802 | 0.886 | 0.936 |
| GENRET (+QG) | 0.681 | *0.861* | 0.888 | 0.952 | 0.759 | *0.789* | *0.881* | *0.940* |
| **Ours: Hierarchy-aware trained dense retrievers** | | | | | | | | |
| HCE-J (`base`) | 0.620 | 0.876 | 0.906 | 0.969 | 0.705 | 0.764 | 0.881 | 0.946 |
| HCE-J (`base`), +QG | 0.640 | 0.878 | 0.914 | 0.973 | 0.742 | 0.785 | 0.898 | 0.954 |
| HCE-J (`large`) | 0.695 | 0.899 | 0.933 | 0.976 | 0.789 | 0.768 | 0.890 | 0.956 |
| HCE-J (`large`), +QG | **0.712** | **0.906** | **0.939** | **0.979** | **0.801** | **0.803** | **0.906** | **0.962** |
| *Hierarchy co-training removed* | 0.700 | 0.899 | 0.931 | 0.977 | 0.793 | 0.783 | 0.894 | 0.957 |

Table 1: Performance on NQ320k and TriviaQA. Numbers typeset in italics are recomputed if model output is public; or rerun from their public implementations. Missing numbers due to our inability to reproduce the exact method to our best effort (may be due to unreleased source code). All rows labeled with `base`/`large` are based on T5-`base`/`large`, thus sharing the same number of parameters for fair comparison.

**Discussion: Effect of Tree Depth** A critical hyperparameter of HCE is $b$, the *branching factor* that controls the tree depth and the expected number of children for each intermediate node. Intuitively, the deeper the hierarchy is, the finer-grained tiered negative samples each relevant document receives.

We ran experiments for NQ320k under the best setup (HCE-J, `large`, +QG), with $b \in \{2, 4, 8, 16, 32, 64, 128, 256\}$. Recall at $\{1, 5, 10, 100\}$ with varying $b$ are plotted in Figure 4, with the tree depth also shown ($L = \lceil \log_b |\mathcal{D}| \rceil$). We set $M = L - 1$: the last layer is computed via negative sampling.

It can be seen that recall at 10/100 remains mostly the same for all $b$. A significant difference can be observed for recall@1: The best result, $b = 8$, has R@1 = 0.712, whereas for $b \geq 64$, R@1 $\approx 0.67$. This 4% gap can be attributed to the finer-grained hierarchy: the sampled negative siblings are much closer to the positive document, generating harder negative samples for contrastive learning.

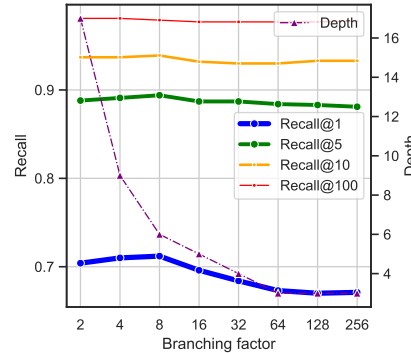

Figure 4: Recall@$\{1, 5, 10, 100\}$ for various branching factors $b$ under NQ320k.

## 5.2 UNSUPERVISED SETTING: BEIR

We evaluate on the BEIR benchmark (Thakur et al., 2021) that contains a diverse set of IR tasks. We use a subset (BEIR-14) that consists of 14 of these datasets following prior work in Izacard et al. (2022): MS MARCO (Nguyen et al., 2016), which is already used as a fine-tuning dataset in our initial checkpoint GTR, and also datasets without a public license, are excluded.

The branching factor $b$ is universally set to be the smallest power of 2 such that the hierarchy depth $L = 3$, i.e., $\tilde{b} = \sqrt[3]{|\mathcal{D}|}$; $b = 2^{\lceil \log_2 \tilde{b} \rceil}$. For example, given a document set of 1M documents, we have

| Dataset | BM25 | DPR | ANCE | TAS-B | GENRET +QG | T5-base | | T5-large | |
|---|---|---|---|---|---|---|---|---|---|
| | | | | | | GTR | HCE-U | GTR | HCE-U |
| TREC-COVID | 0.656 | 0.332 | 0.654 | 0.481 | 0.718 | 0.589 | 0.688 | 0.591 | **0.724** |
| NFCorpus | 0.325 | 0.189 | 0.237 | **0.383** | 0.316 | 0.304 | 0.328 | 0.329 | 0.349 |
| NQ | 0.329 | 0.474 | 0.446 | 0.463 | | 0.495 | 0.514 | 0.547 | **0.561** |
| HotpotQA | **0.603** | 0.391 | 0.456 | 0.584 | | 0.535 | 0.567 | 0.579 | 0.590 |
| FiQA-2018 | 0.236 | 0.112 | 0.295 | 0.300 | 0.302 | 0.352 | 0.388 | 0.424 | **0.473** |
| ArguAna | 0.315 | 0.175 | 0.415 | **0.429** | 0.343 | 0.363 | 0.374 | 0.380 | 0.387 |
| Touché-2020 | **0.367** | 0.131 | 0.240 | 0.162 | | 0.303 | 0.321 | 0.326 | 0.339 |
| CQADupStack | 0.299 | 0.153 | 0.296 | 0.314 | | 0.118 | 0.141 | 0.127 | **0.153** |
| Quora | 0.789 | 0.248 | 0.852 | 0.835 | | 0.880 | 0.874 | **0.885** | 0.880 |
| DBPedia-entity | 0.313 | 0.263 | 0.281 | 0.384 | | 0.347 | 0.358 | 0.391 | **0.408** |
| SciDocs | 0.158 | 0.077 | 0.122 | 0.149 | 0.149 | 0.140 | 0.160 | 0.158 | **0.183** |
| FEVER | **0.753** | 0.562 | 0.669 | 0.700 | | 0.646 | 0.671 | 0.682 | 0.714 |
| Climate-FEVER | 0.213 | 0.148 | 0.198 | 0.228 | | 0.241 | 0.269 | 0.262 | **0.277** |
| SciFact | 0.665 | 0.318 | 0.507 | 0.643 | 0.639 | 0.595 | 0.640 | 0.631 | **0.674** |
| Average (<200k) | 0.393 | 0.201 | 0.372 | 0.398 | 0.411 | 0.391 | 0.430 | 0.420 | **0.465** |
| Average | 0.430 | 0.255 | 0.405 | 0.432 | | 0.422 | 0.450 | 0.451 | **0.479** |

Table 2: NDCG@10 for BEIR-14 datasets under unsupervised training.

| Dataset | T5-base | | | | T5-large | | | |
|---|---|---|---|---|---|---|---|---|
| | GTR | HCE-U | HCE-S | HCE-J | GTR | HCE-U | HCE-S | HCE-J |
| NFCorpus | 0.304 | 0.328 | **0.337** | 0.331 | 0.329 | 0.349 | **0.423** | 0.403 |
| HotpotQA | 0.535 | 0.567 | 0.595 | **0.605** | 0.579 | 0.590 | 0.663 | **0.676** |
| FiQA-2018 | 0.352 | 0.388 | 0.392 | **0.401** | 0.424 | 0.473 | 0.482 | **0.491** |
| SciFact | 0.595 | 0.640 | 0.746 | **0.764** | 0.631 | 0.674 | 0.805 | **0.811** |

Table 3: NDCG@10 for BEIR datasets with a training set under different training scenarios.

$\tilde{b} = \sqrt[3]{1000000} = 100$, hence we set $b = 128$. In the HCE loss, we choose $M = 2$, i.e., the loss of the first two layers are computed via the hierarchy, whereas the last year through contrastive sampling.

**Baselines and Metrics**  BEIR is designed for *zero-shot retrieval*: as such, we compare our unsupervised HCE-U with various baselines. Baselines include BM25, dense retrievers DPR, ANCE, GTR, and TAS-B (Hofstätter et al., 2021), a dense retriever trained with balanced topic aware sampling. Additionally, GENRET, the state-of-the-art generative retriever, reported its performance on the 6 BEIR datasets whose corpus has <200k documents. [10] Following the setup in BEIR, the preferred metric is NDCG@10 and recall@100 (R@100).

**Results**  Unsupervised HCE consistently outperform GTR (with the exception of Quora), with an improvement of $\approx$ 1–11% (on average 3%) for NDCG@10 (Table 2) and $\approx$ 3% for R@100 ( Appendix C). We see that the distribution of the corpus plays an important role here: for corpus like TREC-COVID (biomedical literature) which deviates from the distribution of normal web search text, we see a 10% improvement after unsupervised fine-tuning; for datasets like NQ, HotpotQA, and DBPedia-entity where the documents come from Wikipedia, the performance gain is less, presumably because the pre-trained GTR has already seen text from Wikipedia. Unsupervised fine-tuning does not yield a gain in Quora, because the documents being retrieved are also queries (the task asks the model to find potentially duplicate questions asked on Quora), which are short — the inverse cloze task (ICT) sampling that takes a short excerpt does not work properly in this case.

**Discussion: Effect of Supervision**  We study the effect of (un)supervised HCE training. We take the subset of BEIR where training data is provided publicly, and run HCE under 3 scenarios: unsupervised, supervised, and joint training. Results are reported in Table 3.

We see that with supervision, adding unsupervised training as an auxiliary task results in $\approx$1% gain in NDCG@10, showing that the joint loss in Equation 7 is effective. The performance drop in NFCorpus is due to the dataset's imbalanced training set.[11]

**Discussion: Scalability and Complexity for Clustering**  At training time, apart from the encoder, the centroids of the first $M$ layers of the hierarchy are optimized as parameters, but can be discarded at

---

[10] We found that GENRET does not scale to corpora larger than 200k documents owing to memory issues.

[11] NFCorpus has more than 100k training pairs but only 3k documents in the corpus — setting the loss weight $\alpha = 1$ (balanced between supervised and unsupervised objective) may be detrimental here.

indexing and retrieval time. This requires $O(b^M n)$ extra space at training, where $n$ is the dimensionality of the embeddings. In practice, with a corpus with $N \approx 5,000,000$ documents (e.g. Wikipedia articles), we have $b = 256, M = 2, n = 768$. This space consumption is tolerable under this scenario.

The $k$-means algorithm runs in $O(INkn)$ time, where $I$ is the number of iterations. We observed in all datasets the iteration process converged before $I < 256$. Considering $I$ a constant, the total time complexity for constructing a document hierarchy is $\sum_{i=1}^{\lceil \log_b N \rceil} O\left(\frac{N}{b^{i-1}} \cdot \frac{N}{b^i} \cdot n\right) \simeq O(N^2 n/b)$. This complexity is high, but ameliorated since it can be parallelized on GPUs, and it can be made to a streaming version for lower memory consumption.[12]

### 5.3 INCREMENTALLY UPDATED RETRIEVAL

Our proposed HCE supports on-the-fly update of documents to the index without training, since the new documents can just be encoded and added to the index. However, if too many documents are added to the extent that the corpus distribution is distorted, a re-training and re-indexing might be needed. We study this scenario of a streaming document set under the setting of IncDSI (Kishore et al., 2023), who proposed the setting of *incrementally updated retrieval*. A dataset $\mathcal{D}$ is partitioned into 3 subsets: the old set $\mathcal{D}_0$ (90%) which is used to train the retriever; the new set $\mathcal{D}'$ (9%), in a streaming fashion after training; and a tuning set $\mathcal{D}^*$ (1%) for validation purposes.

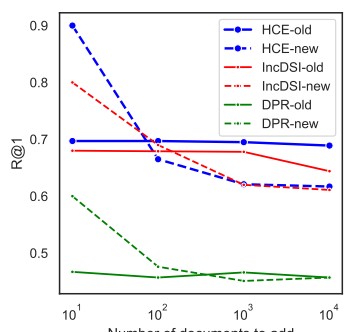

We use the dataset partitions for NQ320k provided in Kishore et al. (2023). Following their setup, we add $k \in \{10^1, 10^2, 10^3, 10^4\}$ new documents in $\mathcal{D}'$ to an index trained with the same old set $\mathcal{D}_0$. Since our method of adding new documents simply involves encoding the new documents, there is no parameter to tune so $\mathcal{D}^*$ is discarded. We measure R@1 and R@10 at different $k$ (number of documents added) for both the original test set in $\mathcal{D}_0$ and the queries associated with the new documents in $\mathcal{D}'$ (denoted "old/new" in Figure 5).

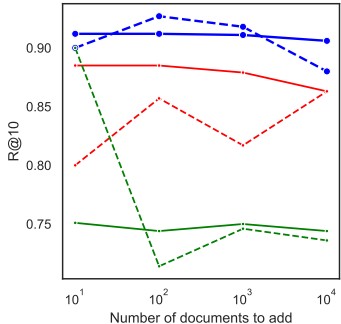

We take the results reported in Kishore et al. (2023), including methods that do not require long-time post-training after adding new documents. These include IncDSI, a constrained optimizer that finds a new embedding for a new document in the embedding space of *an atomic DSI* (§3); and DPR, a contrastively-trained retriever. [13]

Figure 5: Recall@$\{1, 10\}$ for incremental updates on NQ320k.

Results are shown in Figure 5. We see from the trends that all methods' performance dropped for new documents as expected (especially when $k$ goes from $10^2$ to $10^4$), but HCE's performance drop for new documents is comparable to IncDSI, a method specifically designed for incremental additions. At $k \in \{10^3, 10^4\}$, HCE performs comparably with IncDSI under R@1 for new documents in $\mathcal{D}'$, and significantly outperforms under R@10.

## 6 CONCLUSION

We present *hierarchical corpus encoder* (HCE) that jointly learns a dense encoder and a document hierarchy for information retrieval. HCE contrasts positive samples against siblings nodes on the document hierarchy as negative samples, mimicking the training dynamics in hierarchical generative retrieval. HCE is shown to achieve superior performance over a variety of dense encoder and generative retrieval baselines, under both supervised and unsupervised scenarios, demonstrating the effectiveness of jointly learning a document hierarchy.

HCE easily scales to corpora with >5M documents. Additionally, as a dense encoder, HCE supports easy addition and removal of documents to an MIPS index without the need for continued training, as is demonstrated in our incrementally updated retrieval experiments.

---

[12] Our method runs easily on the whole Wikipedia as corpus, with >5M documents to be indexed.

[13] Note that these are *not* directly comparable since IncDSI/DPR are based on BERT while HCE is on T5.

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

## A    DETAILS OF THE CONTRASTIVE LOSS

The contrastive loss employed utilizes both the *in-batch negative trick* and *bidirection contrastive loss*. Mathematically it can be written as

$$
\mathcal{L}_C^{\leftrightarrow}(B) = -\frac{1}{|B|} \sum_{i \in B} \left[ \underbrace{\log \frac{\exp S(q_i, d_i^+)}{\sum_{j \in B} \exp S(q_i, d_j^+) + \sum_{d_i^- \in D_i^-} \exp S(q_i, d_i^-)}}_{\text{query to doc}} + \underbrace{\log \frac{\exp S(q_i, d_i^+)}{\sum_{j \in B} \exp S(q_j, d_i^+)}}_{\text{doc to query (no neg. samples)}} \right]. \quad (8)
$$

## B    STATISTICS OF THE DATASETS

Statistics of the datasets used is provided for reference. Note that the "NQ" dataset in BEIR is a differently preprocessed version of NaturalQuestions (Kwiatkowski et al., 2019) than NQ320k.

| Dataset | # Training pairs | # Test queries | # docs in corpus |
|---|---|---|---|
| NQ320k | 152,144$^{\dagger}$ | 7,830 | 109,739 |
| TriviaQA | 110,647 | 7,701 | 73,970 |
| **BEIR** | | | |
| TREC-COVID | - | 50 | 171332 |
| NFCorpus | 110,575 | 323 | 3633 |
| NQ | - | 3452 | 2681468 |
| HotpotQA | 170,000 | 7,405 | 5,233,329 |
| FiQA-2018 | 14,166 | 648 | 57,638 |
| ArguAna | - | 1,406 | 8,674 |
| Touch'e-2020 | - | 49 | 382,545 |
| CQADupStack | - | 13,145 | 457,199 |
| Quora | - | 10,000 | 522,931 |
| DBPedia-entity | - | 400 | 4,635,922 |
| SciDocs | - | 1,000 | 25,657 |
| FEVER | - | 6,666 | 5,416,568 |
| Climate-FEVER | - | 1,535 | 5,416,593 |
| SciFact | 920 | 300 | 5,183 |

Table 4: Statistics on various datasets used. $^{\dagger}$ Negative pairs are discarded.

# C  ADDITIONAL METRICS OF BEIR

For zero-shot retrieval in BEIR, prior work also reported recall@100 (R@100). We report the performance of our unsupervised model (HCE-U) here, as compared to various baselines. GENRET does not report R@100.

| Dataset | BM25 | DPR | ANCE | TAS-B | T5-base | | T5-large | |
|---|---|---|---|---|---|---|---|---|
| | | | | | GTR | HCE-U | GTR | HCE-U |
| TREC-COVID | **0.498** | 0.457 | 0.457 | 0.387 | 0.411 | 0.444 | 0.434 | 0.457 |
| NFCorpus | 0.250 | 0.208 | 0.232 | 0.280 | 0.275 | 0.305 | 0.298 | **0.334** |
| NQ | 0.760 | 0.880 | 0.836 | 0.903 | 0.893 | 0.918 | 0.930 | **0.945** |
| HotpotQA | **0.740** | 0.591 | 0.578 | 0.728 | 0.676 | 0.704 | 0.725 | 0.739 |
| FiQA-2018 | 0.539 | 0.342 | 0.581 | 0.593 | 0.670 | 0.702 | 0.734 | **0.789** |
| ArguAna | 0.942 | 0.751 | 0.937 | 0.942 | 0.974 | 0.976 | 0.978 | **0.979** |
| Touché-2020 | **0.538** | 0.301 | 0.458 | 0.431 | 0.488 | 0.510 | 0.500 | 0.519 |
| CQADupStack | 0.606 | 0.403 | 0.579 | 0.622 | 0.221 | 0.251 | 0.234 | **0.265** |
| Quora | 0.973 | 0.470 | 0.987 | 0.986 | 0.994 | 0.994 | **0.995** | 0.995 |
| DBPedia-entity | 0.398 | 0.365 | 0.319 | 0.499 | 0.418 | 0.440 | 0.480 | **0.511** |
| SciDocs | 0.356 | 0.360 | 0.269 | 0.335 | 0.319 | 0.384 | 0.354 | **0.429** |
| FEVER | 0.931 | 0.840 | 0.900 | 0.937 | 0.923 | 0.935 | 0.941 | **0.947** |
| Climate-FEVER | 0.436 | 0.427 | 0.445 | 0.534 | 0.522 | 0.565 | 0.552 | **0.582** |
| SciFact | 0.908 | 0.727 | 0.816 | 0.891 | 0.860 | 0.904 | 0.894 | **0.932** |
| Average | 0.634 | 0.509 | 0.600 | 0.648 | 0.617 | 0.645 | 0.646 | **0.673** |

Table 5: Recall@100 for BEIR-14 datasets under unsupervised training.

