# OpenReview forum: "Hierarchical Corpus Encoder: Fusing Generative Retrieval and Dense Indices"
_ICLR.cc/2025/Conference — ICLR 2025 Conference Withdrawn Submission_

### Official Review · Reviewer_WDvU · 2024-11-02

**Soundness:** 3
**Presentation:** 2
**Contribution:** 2
**Rating:** 3
**Confidence:** 5

**Summary:**

The authors propose a hierarchical dense retrieval (not generative retrieval) method (***This claim is crucial because the paper's writing from a generative retrieval perspective can be misleading***). Since it is hierarchical, it is essential to consider the source of the documents' hierarchical information. In this paper, the hierarchical structure of the documents is derived from the distances between the initial document embeddings, representing an implicit form of hierarchical information.

**Strengths:**

I believe the proposed method is versatile, as it combines dense retrieval with implicit hierarchical information without relying on externally defined document relationships.

**Weaknesses:**

1. The writing logic of this paper is highly misleading. The authors begin by discussing generative retrieval, claiming that its drawback is the inability to update documents. However, the method they propose is actually a dense retrieval approach, and they claim their method supports document updating and zero-shot retrieval as advantages. However, any dense retrieval method naturally supports document updates. ***If the logic of this paper were valid, anyone developing a dense retrieval method in the future could simply use the limitations of generative retrieval as motivation for their work.***

2. Since the authors essentially propose a hierarchical dense retrieval method, it is important to introduce and compare it with existing dense retrieval methods that learn hierarchical information. Unfortunately, the paper completely skips this most relevant part.

    2.1. First, credit should be given to previous hierarchical dense retrieval methods, e.g., [1], including a related work section to provide an overview.

    2.2. The paper should explain how their approach differs from previous hierarchical dense retrieval methods and where it might offer advantages.

    2.3. A fair comparison with previous hierarchical dense retrieval methods should be conducted in the experiments.

3. Compared to previous hierarchical dense retrieval methods, I do not find this paper to have sufficient novelty. Perhaps this is why the authors chose to skip related work on hierarchical dense retrieval and instead writing from generative retrieval.

[1] Liu, Y., Hashimoto, K., Zhou, Y., Yavuz, S., Xiong, C., & Yu, P. S. (2021). Dense hierarchical retrieval for open-domain question answering. arXiv preprint arXiv:2110.15439.

**Questions:**

1. First, I hope the authors address the weaknesses mentioned above.
2. “Generative retrieval employs sequence models for conditional generation of document IDs based on a query (DSI (Tay et al., 2022); NCI (Wang et al., 2022); inter alia). While this has led to improved performance in zero-shot retrieval …”. The claim that generative retrieval leads to improved performance in zero-shot retrieval is inaccurate or even entirely incorrect. Both DSI and NCI require training to achieve any level of effectiveness; without training, their performance would be negligible. As far as I understand, generative retrieval fundamentally relies on training and does not inherently address the challenges of zero-shot retrieval.
3. “better performance in unsupervised settings when adapted to a new corpus without any labeled training data.” It is unclear which previous work has substantiated this assertion. I am carious how the authors conclude this claim from the previous generative retrieval works.
4. “Here we examine the innovations of generative retrieval and identify the key important distinction with dense retrieval approaches to date: tiered hierarchical negative sample”. Could the author provide more explanation and motivation in the introduction? Presenting just this term without any explanation of its meaning might leave readers confused.
5. “These embeddings are updated under gradient descent for every training iteration. This is different from contrastive learning in dense retrievers, where usually a small set of negative samples are sampled from the corpus.” This claim is incorrect. There is no fundamental difference between the automatic version and dense retrieval. The atomic version can train document embeddings only because the dataset used, such as NQ320K, has a relatively small number of documents. In this case, all documents other than the positive ones can be considered as negative examples. However, when the number of documents increases, negative sampling becomes necessary.
6. Please include a related work section, particularly focusing on previous work related to hierarchical dense retrieval, in future revisions.
7. Please add methods from hierarchical dense retrieval as baselines, as this is crucial for determining whether the proposed method is effective.
8. I suggest that the authors avoid framing the paper from the perspective of generative retrieval. While the work may have been inspired by generative retrieval hierarchy IDs, since the focus is on hierarchical dense retrieval, the writing should be approached from that standpoint.

---

### Official Review · Reviewer_L85d · 2024-11-04

**Soundness:** 3
**Presentation:** 3
**Contribution:** 2
**Rating:** 5
**Confidence:** 4

**Summary:**

This paper proposes a hierarchical corpus encoding method for the dense retriever inspired by their successful applications on generative retrieval.  Starting from an initial encoder, the documents in the corpus are hierarchically clustered in the tree structure. The dense encoder is learned to maximize the distances between clustered nodes and other nodes at each depth using contrastive learning loss. Moreover, the authors leverage the EM-style algorithm to iteratively optimize the clustering and the embedding. Experimental results illustrate obvious improvements in the proposed method over several baselines

**Strengths:**

1. The paper writing is clear and easy to follow.
2. The conducted experiments are adequate with detailed analysis and ablation studies. The experimental results demonstrate obvious improvements over different types of baselines.

**Weaknesses:**

1. The motivation of this paper is not well explained.  The DocIDs built upon a hierarchic document structure are used for sequential generation. In dense retrieval, it is no necessary to construct a hierarchical version of encoding.  Besides, if the generative retrieval suffers from the generalization problem when adding new documents, why the proposed method can do better? Adding more documents may destroy the tree structure of previous documents, making it an unreasonable design for dense retrieval.
2. The contribution is incremental since both ideas of hierarchical clustering and iteratively embedding learning have been adopted in previous studies.
3. The Co-Training of Encoder and Hierarchy part is not detailed enough, lacking a detailed description of the training process and convergence proof.

**Questions:**

Please add detailed introduction and convergence proof in the Co-Training of Encoder and Hierarchy part

---

### Official Review · Reviewer_t4bz · 2024-11-06

**Soundness:** 2
**Presentation:** 2
**Contribution:** 2
**Rating:** 3
**Confidence:** 4

**Summary:**

This paper explores generative retrieval, which uses sequence models to generate document IDs based on a query, as seen in methods like and NCI. While this approach has demonstrated improvements in zero-shot retrieval, a major challenge remains in supporting documents that were not part of the training data. The authors argue that the effectiveness of generative retrieval is rooted in contrastive training between sibling nodes in a document hierarchy. To address this, they propose the Hierarchical Corpus Encoder, which can be paired with traditional dense encoders. Through experiments, they demonstrate that HCE outperforms generative retrieval models in both unsupervised zero-shot and supervised settings. However, the novelty of the motivation behind the approach is somewhat limited, as the concept of hierarchical encoding has been explored in other contexts. Overall, the current version of the manuscript is not sufficient for publication at a top-tier conference like ICLR.

**Strengths:**

1.	This manuscript identifies the performance of generative retrieval lies in contrastive training between sibling nodes in a document hierarchy.
2.	And it proposes, the hierarchical corpus encoder (HCE), which can be supported by traditional dense encoders.
3.	The experiments show that HCE achieves superior results than generative retrieval models under both unsupervised zero-shot and supervised settings, while also allowing the easy addition and removal of documents to the index.

**Weaknesses:**

1. Analyzing the role of contrastive learning in the generative retrieval framework: This perspective is not novel and can be referenced from [1]. This manuscript proposes combining docid encoding with model learning, which seems intuitive; however, this is almost the only innovative point of this work.

   [1] Generative Retrieval as Multi-Vector Dense Retrieval

2. Regarding the experiments: The baselines used for comparison are too limited, making the effectiveness of this work unconvincing. Common baselines in generative retrieval studies, such as NOVO [2], Ultron [3], LTRGR [4], GenRRL [5], RIPOR [6], ListGR [7], SE-DSI [8], etc., should be included. Moreover, the dense retrieval baselines are not state-of-the-art; adding a new baseline, such as [9], is recommended. [9] has been made publicly available, making it easy to conduct experiments.

   [2] Novo: Learnable and interpretable document identifiers for model-based IR
   [3] An ultimate retriever on corpus with a model-based indexer
   [4] Learning to rank in generative retrieval
   [5] Enhancing generative retrieval with reinforcement learning from relevance feedback
   [6] A neural corpus indexer for document retrieval
   [7] Listwise Generative Retrieval Models via a Sequential Learning Process
   [8] Semantic-Enhanced Differentiable Search Index Inspired by Learning Strategies
   [9] Fine-Tuning LLaMA for Multi-Stage Text Retrieval

3. Additionally, the document count in this work’s datasets is relatively small. In RIPOR [6], a dataset with millions of documents was explored, and it would be highly interesting to see how this method performs on medium-to-large-scale datasets.

**Questions:**

See the weaknesses above.

---

### Official Review · Reviewer_8Pr5 · 2024-11-06

**Soundness:** 3
**Presentation:** 3
**Contribution:** 3
**Rating:** 6
**Confidence:** 4

**Summary:**

The paper presents the Hierarchical Corpus Encoder (HCE), a new approach in information retrieval that combines aspects of generative retrieval and dense indices for efficient and flexible document retrieval. Traditional generative retrieval methods struggle with the addition of new documents and are limited by computational constraints, while dense encoders often require external indices, which can be less adaptable in zero-shot or dynamic environments. HCE addresses these challenges by introducing a hierarchical structure to organize documents, enabling contrastive training with “sibling” negative samples from within this hierarchy. This allows for efficient nearest-neighbor search and avoids the retraining issues seen in generative retrieval models. The experimental results demonstrate that HCE surpasses both dense and generative retrieval baselines in zero-shot and supervised scenarios, making it a promising solution for scalable and adaptable document retrieval.

**Strengths:**

- The writing is easy to follow and the logic is clear.
- HCE creatively integrates the strengths of generative and dense retrieval, leveraging a hierarchical structure for efficient storage and retrieval. This fusion allows for flexible document indexing, making it possible to add or remove documents without extensive retraining, which is often a limitation in other retrieval methods.
- The authors provide extensive experimental results across both unsupervised zero-shot and supervised settings , validating the robustness and superiority of HCE over state-of-the-art approaches.

**Weaknesses:**

- The hierarchical clustering process, especially as described in HCE, may introduce additional computational overhead during initial setup. While this step is not frequently repeated, it could be challenging for very large corpora or low-resource environments that cannot afford extensive preprocessing.

**Questions:**

- Why generative retreival method get better performance when adapted to a new corpus without any labeled training data? I mean when the model didn’t see the identifier before, it also can not get a good performance.
- How does HCE handle out-of-vocabulary or new domains in zero-shot settings?

---

### Note · Authors · 2024-11-24

I have read and agree with the venue's withdrawal policy on behalf of myself and my co-authors.